# Clarifying the structure of serious head and spine injury in youth Rugby Union players

Koh Sasaki[1]*, Haruhiko Sato[2], Akihiko Nakamura[3], Takumi Yamamoto[4], Ichiro Watanabe[5], Takashi Katsuta[6], Ichiro Kono[7]

1 Research Center of Health, Physical Fitness, and Sports, Nagoya University, Nagoya, Japan, 2 Department of Neurosurgery, Seirei Mikatahara General Hospital, Shizuoka, Japan, 3 Department of Pediatric Surgery, Nakamura Hospital, Tokyo, Japan, 4 Faculty of Education, National Defense Academy, Yokosuka, Japan, 5 Faculty of Liberal Arts, Tokyo City University, Tokyo, Japan, 6 High Performance Sport Center, Japan Sport Council, Tokyo, Japan, 7 2019 Rugby World Cup Organizing Committee, Tokyo, Japan

* sasakikoh@htc.nagoya-u.ac.jp

**Data Availability Statement:** All relevant data are within the manuscript and its Supporting Information files.

## Abstract

This study aimed to clarify the cause of rugby head and spinal cord injuries through a network centrality analysis of 14-year (2004–2018) longitudinal data in Japan. The study hypothesis is that understanding the causal relationship among the occurrence of serious injuries, the quality of player experience and play situation as a network structure could be possible to obtain practical knowledge on injury prevention. In this study, bipartite graphs are used to make it easier to understand the situation of players and injuries. This would also help to elucidate more characteristic subgroup. A network bipartite graph and subgroup (cluster) analyses were performed to clarify the injured players' experience and the cause of injury. We used the algorithm of R program, IGRAPH, clustering edge betweenness. For subgroup extraction, the modularity Q value was used to determine which step to cut. The Japanese rugby population was 93,873 (2014–2018 average), and 27% were high school students. The data showed that careful attention would be particularly needed for groups of inexperienced Japanese high school players. Our study suggests that we should consider introducing rules that prohibit "head-on collisions" in youth rugby.

## Introduction

In Japan, approximately 20 cases of serious injuries occurred annually since 2011 [1]. Data utilizing long-term data would contribute to its prevention. In rugby, tackle plays to stop the opponent's attack by physical contact are allowed by rules. This could lead to more serious injuries in adolescents, where the growth gap is large. A longitudinal study of adolescent injuries may be an important issue. However, studies on serious rugby injuries are few, which are mostly short-term or conceptual ones [2,3,4], and longitudinal studies on these cases are scarce. The reason why there are few longitudinal studies would be that the social rules for ethical use of medical diagnosis and for protecting personal information were not sufficiently established.

The occurrence of injury will have a relationship structure, such as experience differences and growth differences, as well as physical contact situations. The approach of grasping the

**Funding:** Initials of the authors who received JSPS KAKENHI grant Number 16K01658(2016-2018) and 19K11549(2019-2021).

**Competing interests:** The authors have declared that no competing interests exist.

network structure of its occurrence might be useful for safety management of tackle situations in Rugby.

Network analysis has applications in many disciplines including physics, computer science, biology, economics, ecology, and sociology. From a broad perspective, networks can be viewed as integrative structures of the physical society or lifeforce. Additionally, the relationship that links structural entities is always dynamic. Network analysis was found to be a valuable new tool for exploring, depicting, and explaining the individual relationships that impact team dynamics [5]. Studying these collaborative structures might allow discovery of the possibilities of new survival abilities in a society.

Network analysis has been developed in the field of communication network studies [6,7].

Some traditional studies focusing on network structures were introduced in the period from 1960 to 1970, including the "small world phenomenon", "the strength of weak ties", and "scale free network". In addition, the specification of the "cluster community" or "sub-group network" were also developed to determine with social network robustness. Sociological approaches are widely used for predictive models of big data in the field of information technology and cerebral functions in biology [8,9,10].

Network analysis methodologies have been applied in social and natural science approaches used in sport sciences. Within the social-science approaches, there are social psychological debates on how sports assist the development of cooperation values within a specific society [6,11]. In addition, political and economic studies have been reported that declining trends in sports and health promotion activities have a negative impact on the civil economy network [12,13,14,15]. Natural science approaches include discussions on the causes of sports injuries [4,13]. The factors that play a central role in various physiological parameters during exercise-induced fatigue were clarified by network analysis, and its application to risk management was discussed [13]. The direct relationship among some performance indicators in sports had also been investigated [14,15,16,17,18,19], which involves strategies and tactics in competitive dimensions. It was considered that there was a central player called a hub in the passing behavior research of a soccer game, and there was a power law there. Furthermore, the hub dynamically switched throughout the game. The difference between dynamic networks and static networks is that the former focuses on the variable and diverse play structures occurring in sports games [18].

How is the importance of network dynamism discussed? There exist important hub functions, sub-group communities, and actors playing some central roles within a network. The important role of network analysis would be to reveal the decisive and pragmatic structures required to obtain specific goals among complex networks as well as to identify individuals who are fulfilling the principal roles. This could be discussed from the perspective of network centrality [20,21]. Network centrality clarifies an organization's temporal and bipartite or multilayered structures, which would help us understand the driving force of the network dynamism [22].

This study aimed to investigate the network relationship of injury cause structure in a sports field by clarifying the situation of serious injury by network centrality analysis. The study hypothesis is that understanding the causal relationship among the occurrence of serious injuries, the quality of player experience and the play situation as a network structure, it could be possible to obtain more practical knowledge on injury prevention.

## Materials and methods

### Data collection

The data were obtained from rugby serious injury reports for the past 14 years [1]. Registered teams of the Japan Rugby Football Union are obligated to report within 3 days, 2 months, and

6 months after a serious injury. Multiple specialist doctors determine the classification of the injury: 1) death, 2) loss of consciousness over 24 hours, 3) spinal cord injury with quadriplegia, 4) craniotomy and spinal surgery, 5) visceral injury surgery, and 6) seriously injured from medical certificate. The referring data of the Japan Rugby Union serious injury report were fully anonymized. The anonymized data were locked in multiple stages in a hard disk drive that was not connected to the internet. All procedures used in this study were approved by the Ethics Committee of the Research Center for Health, Physical Fitness, and Sports, Nagoya University.

In this study, high school players were divided into the following groups according to their age and years of experience: 16E1 (players aged 16 years with 1-year rugby experience), 17E2 (players aged 17 years with 1-2-year experience), 18E3 (players aged 18 years with 1-3-year experience), 16EM (players aged 16 years with many years of experience), and 17EM (players aged 17 years with many years of experience), 18EM (players aged 18 years with many years of experience).

Caused of injuries were categorized into 1) own tackle, 2) oppose tackle, 3) rack, a one-on-one situation; 4) collision not categorized as above, 5) saving, hold the ball on the ground, or 6) others or unknown. In addition, the causes of injury were further classified as mauls (3 or more players battling the ball) and scrum (8 players battling the ball). Moreover, seasons were categorized as summer training camp (August) and the main official season (November, February). These classification processes were performed by the safety management committee of the Japanese Rugby Football Union. This committee works closely with the medical committee, coaching committee, referee committee, and technical committee in Japanese Rugby Union, cooperates in classification of the cause of injury based on the detailed play situation reported by the team manager and doctor.

## Network analysis and statistical analysis

Nodes of network in this study are player groups (age and years of experience), types of injuries, and play styles that cause injuries (tackle, tackled, scrum, etc.). It clarifies the continuity of what kind of players' experience years, what kind of play, and what part of body collision caused the injury. These are the dynamic networks which focus on the diverse play structures occurring throughout rugby game. Although this study would clarify a kind of local properties of networks, it might be considered as global properties by promoting international joint study in the future.

A bipartite graph divides a network into two subsets and has no edges between nodes in each set [23]. In this study, bipartite graphs are used to make it easier to understand the situation of players and injuries. Map layout was calculated by Fruchterman-Reingold Algorithm which is a force-directed layout algorithm for centralization of the injury factors [24]. This would also help to elucidate more characteristic subgroup. We used the algorithm of R program, IGRAPH, clustering edge betweenness. The idea of the edge betweenness based community structure detection is that edges connecting separate modules have high edge betweenness as all the shortest paths from one module to another must traverse through them [25,26]. Edge-betweenness centrality represents the degree of being located in the shortest path bridge between nodes. It was adopted because it would indicate a close relationship among players, injuries, and the causes. The limitation of this metric is that its approximate value may become unstable in case of a large-scale network graph [27,28].

In the bipartite graphs, players' experiences, injury details, and caused plays were plotted in the adjacent matrix. For drawing the graph, centrality, obtained by identifying which items occupy the critical positions in injury condition, is a major focus. Subgroup that was central in

the network area are extracted from the relatively dense area. If the adjacent matrix of the graph is $A = (\alpha_{ij})$, the number of nodes of the entire graph is $n$, the number of nodes belonging the subgroup is $n_s$, and the cohesion $S$ of the subgroup is next [10].

$$S = \frac{\sum_{i \in S} \sum_{j \in S} a_{ij}}{n_s(n_s - 1)} \Big/ \frac{\sum_{i \in S} \sum_{j \notin S} a_{ij}}{n_s(n - n_s)}$$

There were some clustering methods that extract subgroup such as random walks [29], greedy algorithm [30], overlapping [31], or spin glass [32]. The present study repeated the calculation of edge-betweenness centrality and the removal of the edge with the maximum betweenness centrality to detect a highly cohesive subgroup. The use of edge-mediated centrality makes it possible to extract a subgroup relatively easily [25,26]

For subgroup extraction, the modularity Q value was used to determine which step to cut. If the extracted subgroup set is $C$, the number of edges connected from community $i$ to community $j$ is $?_{??}$, and the total number of edges included in the entire graph structure is $m$, Modularity is defined as follows [33]:

$$Q = \sum_{i \in C} \{e_{ij}/2m - (\sum_{j \in C} e_{ij}/2m)^{\wedge}2)\}$$

The closer the modularity Q value is to 1, the more appropriate the cluster is divided. There were few graphs close to 1 in this study.

## Results

In the entire domestic rugby population of 93,873 (2004–2018 average), there is a higher percentage (27%) of high school students (~18 years old). Among all high school players, 57% (24,918–10,693 = 14,225) started from high school, indicating that they have a relatively low experience. The serious injury ratio of the high school players was high (head: 41%, spine: 41%, chest abdominal: 50%, circulation: 38%, average: 43%. High school students suffered a relatively high frequency of head injuries during the summer camp in August. This season would be a first time for high school first graders (16 years old) to be suffered of high load practices and games of rugby. Many injuries occurred during the summer camp (August) and the game season (February). (Table 1)

**Table 1. Japanese rugby population, number (ration) of serious injuries, and number of head injuries (HI) and all serious injuries (ASI) by month in High school players (nc; not clear).**

| Japanese Rugby Population | | | | | | | |
|---|---|---|---|---|---|---|---|
| Age grade | -12 | -15 | -18 | -22 | 23- | (women) | Total |
| Number (%) | 19,975 (21) | 10,510(11) | 24.918(27) | 10,693(12) | 26,204(28) | 570 (1) | 93,873(100) |
| Number (Ration) of serious injuries in Japan (nc; not clear) | | | | | | | |
| Head | nc | nc | 48 (41) | nc | nc | nc | 117 |
| Spine | nc | nc | 54 (41) | nc | nc | nc | 133 |
| Chest Abdominal | nc | nc | 4 (50) | nc | nc | nc | 8 |
| Circulation | nc | nc | 10 (38) | nc | nc | nc | 26 |

| Number of head injuries (HI) and all serious injuries (ASI) by month in High school players (nc; not clear) | | | | | | | | | | | | |
|---|---|---|---|---|---|---|---|---|---|---|---|---|
| Month | 4 | 5 | 6 | 7 | 8 | 9 | 10 | 11 | 12 | 1 | 2 | 3 | nc |
| N of HI | 2 | 0 | 5 | 4 | 14 | 3 | 6 | 3 | 2 | 5 | 0 | 3 | 0 |
| N of ASI | 6 | 3 | 2 | 4 | 7 | 4 | 7 | 3 | 2 | 5 | 7 | 3 | 2 |

## Head injury

Table 2 shows that there is a higher occurrence of acute subdural hematoma (ASH) in the high school players. Thirty-five cases (76%) out of the 48 head injuries were occurred in inexperienced players.

The causes of the injuries were own tackle and oppose tackle, accounting for 83% of cases. There were also other collision situations. Table 3 shows the cause of play and the body area of collision. Injury reports suggested that the head position of the injured player tended to be lower than that the opposite's knee height. Many injured parts were at frontal depressions and many situations were in tackles (Table 3).

The ratio of inexperienced players with ASH was high (79%; 22 out of the 28), and the cause play was still due to tackle (Table 2). The number of players who recovered were 13, those with aftereffects were 6, those who died were 2, and those with unknown injuries were 7. The 16E1 group showed unfortunate results.

**Table 2. High school players' head injuries, spinal cord injuries, the age and experiences of playing years (ASH: acute subdural hematoma, FR: fracture, AEH; acute epidural hematoma, ICH/CC: intra cerebral hemorrhage, cerebral contusion, CI/CA: cerebral infarction, cerebrovascular accident O: others or not clear, OTHG: oppose tackle and head to ground, THG; tackle and head to ground, THB: tackle and head to oppose body, C: collision not categorized as above, R: rack, S: saving, VF: vertebral fractureDS: dislocation of spine, FD: fracture dislocation, CC: central cord, S: Spinal cord, CSC: Cervical spinal cord, OT: oppose tackle, T: own tackle, Rack: a one-to-one situation, M: maul (3 or more players battling the ball), C: collision not categorized as above, Saving; hold the ball on the ground, SC: scrum (8 players battling with the ball)).**

| Players | | 16all | 16E1 | 17all | 17E2 | 18all | 18E3 | Total |
|---|---|---|---|---|---|---|---|---|
| head Injuries | ASH | 15 | 12 | 8 | 6 | 5 | 4 | 28 |
| | FR | 2 | 1 | 3 | 1 | 0 | 0 | 5 |
| | AEH | 2 | 1 | 0 | 0 | 3 | 3 | 5 |
| | ICH/CC | 0 | 0 | 1 | 1 | 4 | 2 | 5 |
| | CI/CA | 1 | 0 | 1 | 1 | 0 | 0 | 2 |
| | O | 2 | 2 | 1 | 1 | 0 | 0 | 3 |
| | Total | 22 | 16 | 14 | 10 | 12 | 9 | 48 |
| cause play of ASH | OTHG | 7 | 7 | 3 | 1 | 1 | 1 | 11 |
| | THG | 3 | 2 | 4 | 4 | 2 | 2 | 9 |
| | THB | 3 | 2 | 0 | 0 | 0 | 0 | 3 |
| | C | 2 | 1 | 0 | 0 | 0 | 0 | 2 |
| | R | | | 0 | 0 | 1 | 0 | 1 |
| | S | | | 0 | 0 | 1 | 1 | 1 |
| | O | | | 1 | 1 | 0 | 0 | 1 |
| | Total | 15 | 12 | 8 | 6 | 5 | 4 | 28 |
| Spinal cord injuries | VF | 1 | 1 | 6 | 6 | 6 | 3 | 13 |
| | DS | 1 | 1 | 7 | 5 | 5 | 5 | 13 |
| | FD | 2 | 0 | 6 | 6 | 3 | 3 | 11 |
| | CC | 0 | 0 | 2 | 1 | 4 | 1 | 6 |
| | S | 1 | 1 | 3 | 3 | 0 | 0 | 4 |
| | CSC | 3 | 2 | 0 | 0 | 2 | 2 | 5 |
| | O | 1 | 0 | 1 | 1 | 0 | 0 | 2 |
| | Total | 9 | 7 | 25 | 22 | 20 | 14 | 54 |
| cause play of spinal cord injuries | OT | 2 | 1 | 3 | 2 | 2 | 1 | 7 |
| | T | 2 | 2 | 10 | 10 | 9 | 7 | 21 |
| | R | 2 | 1 | 5 | 4 | 3 | 1 | 10 |
| | M | 0 | 0 | 2 | 2 | 3 | 3 | 5 |
| | C | 0 | 0 | 2 | 1 | 0 | 0 | 2 |
| | S | 1 | 1 | 0 | 0 | 0 | 0 | 1 |
| | SC | 3 | 2 | 3 | 3 | 2 | 2 | 8 |
| | Total | 10 | 6 | 25 | 22 | 19 | 14 | 54 |

**Table 3. Causes of injury and the body area of collision (H-G; head to ground, H-H; head to head, H-L; head to leg, H-B; head to the other part of opponent's body part, LH; low head (head under the opponent's body), O; others, T; own tackle, OT; oppose tackle, R; ruck (a one-on-one situation), C; collision not categorized as above, S; saving (hold the ball on the ground), O; other or unknown, SC; scrum (8 players battling with the ball), M; maul (3 or more players battling the ball), and cause of the load (MW; multi players weights, LH; low head (head under oppose' body), H-B; head to the other part of oppose body, N-B; Neck to other part of oppose body, O: Others).**

| | causes of injury and the body area of collision | | | | | | | cause of the load | | | | | |
|---|---|---|---|---|---|---|---|---|---|---|---|---|---|
| | **H-G** | **H-H** | **H-L** | **H-B** | **LH** | **O** | **Total** | **MW** | **LH** | **H-B** | **N-B** | **O** | **Total** |
| T | 12 | 1 | 0 | 0 | 0 | 0 | 13 | 4 | 7 | 4 | 2 | 4 | 21 |
| OT | 6 | 4 | 5 | 7 | 3 | 2 | 27 | 2 | 0 | 0 | 3 | 2 | 7 |
| R | 0 | 0 | 2 | 0 | 0 | 0 | 2 | 8 | 0 | 0 | 0 | 2 | 10 |
| C | 1 | 2 | 0 | 0 | 0 | 1 | 4 | 0 | 0 | 2 | 0 | 0 | 2 |
| S | 0 | 0 | 1 | 0 | 0 | 0 | 1 | 0 | 0 | 1 | 0 | 0 | 1 |
| O | 0 | 0 | 0 | 0 | 0 | 0 | 1 | | | | | | |
| SC | | | | | | | | 8 | 0 | 0 | 0 | 0 | 8 |
| M | | | | | | | | 5 | 0 | 0 | 0 | 0 | 5 |

## Spinal cord injury

A total of 54 spinal cord injuries occurred in high school students (Table 1). Forty-three players (80%) with spinal cord injuries were inexperienced (Table 2). Causes of spinal cord injuries in the inexperienced players were OT (oppose tackle), T (own tackle), Ruck (a one-on-one situation), Maul (3 or more players battling the ball), C (collision not categorized as above), Saving, and SC (8 players battling with the ball), (Table 3). The cause of the load occurred not only in one-on-one tackles, but also under heavy weights pressure by multiple players, especially in the head and spine, such as scrum or maul (Table 3).

## Junior high and elementary school players

A total of 20 serious injuries were found in elementary and junior high school players (Table 4). The injuries in these age groups should also be prevented the expansion.

In terms of body parts and caused of injuries, there were many cases of head and spinal cord injuries caused by a tackle. There were also many cases of the head hitting the ground.

**Table 4. Serious injuries, injured body parts and caused plays, and the possible sequelae and the collision situations (ASH: acute subdural hematoma, ICH/CC: intra cerebral hemorrhage, cerebral contusion, TSH: traumatic subarachnoid hemorrhage, FR: Fracture, H-G: head to ground, H-H/B: head to head or/and the other part of opposes' body, nc: not clear).**

| | Serious injury | | | | | | | | | | | | Injured body parts and caused plays | | | | | | | | |
|---|---|---|---|---|---|---|---|---|---|---|---|---|---|---|---|---|---|---|---|---|---|
| year | 2007 | 2008 | 2009 | 2010 | 2011 | 2012 | 2013 | 2014 | 2015 | 2016 | 2017 | 2018 | Age (years old) | | | | Caused plays | | | | |
| | | | | | | | | | | | | | -12 | 13 | 14 | 15 | T | OT | R | M | O |
| Head | 2 | 0 | 0 | 0 | 1 | 3 | 1 | 0 | 1 | 0 | 1 | 1 | | 4 | 4 | 3 | 3 | 5 | 1 | | 2 |
| Spine | 0 | 0 | 1 | 0 | 0 | 2 | 0 | 0 | 1 | 0 | 0 | 1 | | 4 | 1 | | 2 | 1 | | 2 | |
| chest abdominal | 0 | 0 | 0 | 0 | 0 | 0 | 0 | 0 | 0 | 0 | 2 | 0 | 1 | | | | 1 | 1 | | | |
| heatstroke | 0 | 0 | 1 | 1 | 0 | 0 | 0 | 0 | 0 | 0 | 0 | 0 | | | | | | | | | |

| Injuries, the possible sequelae and the collision situations. | | | | | |
|---|---|---|---|---|---|
| | possible sequelae | | collision situations. | | |
| | possible sequelae | Total | H-G | H-H/B | nc |
| ASH | 2 | 8 | 3 | | 3 |
| ICH/CC | 1 | 2 | 1 | | |
| TSH | | 1 | 1 | | |
| FR | | 3 | | 3 | |

## Clustering by edge betweenness centrality

Given that the number of serious injury data were not large enough (48 head injuries, 54 spinal cord injuries), we tried to extract the central subgroup in the network. The ratio of the relationship density between the nodes in the subgroup and the nodes outside the subgroup indicates the cohesion of the subgroup. It is a community detection approach based on edge betweenness centrality. The graph is divided from the calculation of the edge-betweenness centrality and the removal of the edge having the maximum betweenness centrality, and a highly cohesive community is detected.

## Head injuries

We made a bipartite graph of age, years of experience, and head injury symptoms (Fig 1). From the edge-betweenness centrality analysis in the network, the central subgroup comprised young and inexperienced players [16E1 (players aged 16 years with 1-year rugby experience), 17E2 (players aged 17 years with 2-year experience), 18E3 (players aged 18 years with 3-year experience), 16EM (players aged 16 years with many years of experience)] suffered serious injuries such as ASH, skull fracture, cerebral infarction/cerebrovascular accident, and others (Fig 1, clustering edge-betweenness, Modularity Q value = 0.5).

A subgroup analysis of causes of serious injuries include own tackle and opponents' tackle situations. The collisions parts were "head to ground", "head to head", 'head to opponent's body", "head under opponent's body" and others. The collision position of the head during

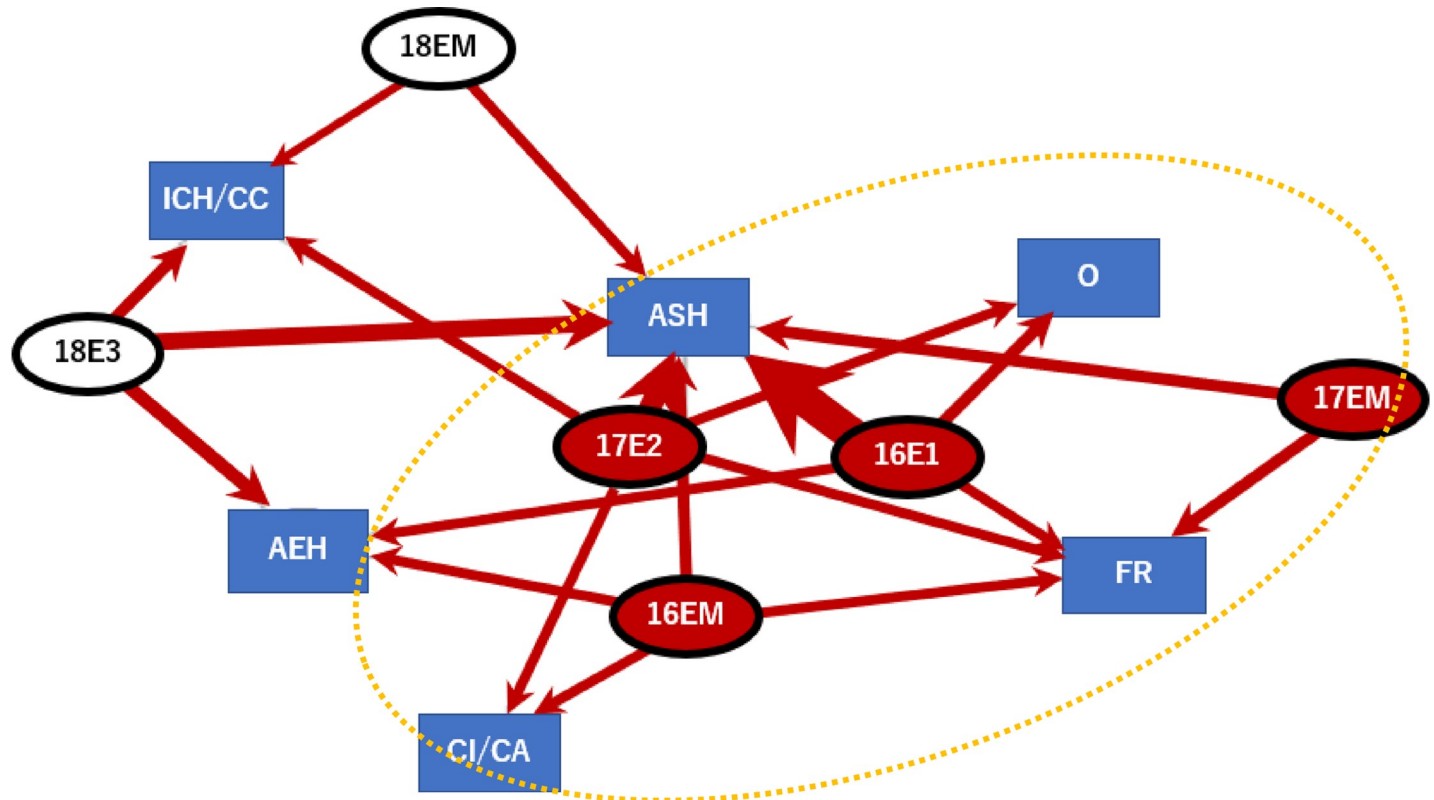

**Fig 1. Age, grade, years of rugby playing experience, presence of head injury [inexperienced players (16 years old and with only 1 year of experience, 16 years old and with many years of experience, 17 years old and with 2 years of experience, 17 years old and with many years of experience), and presence of serious injuries like such as ASH: acute subdural hematoma, FR: skull fracture, CI/CA: cerebral infarction/cerebrovascular accident others (the circle was the extracted subgroup by clustering edge betweenness, Modularity Q value = 0.5)].**

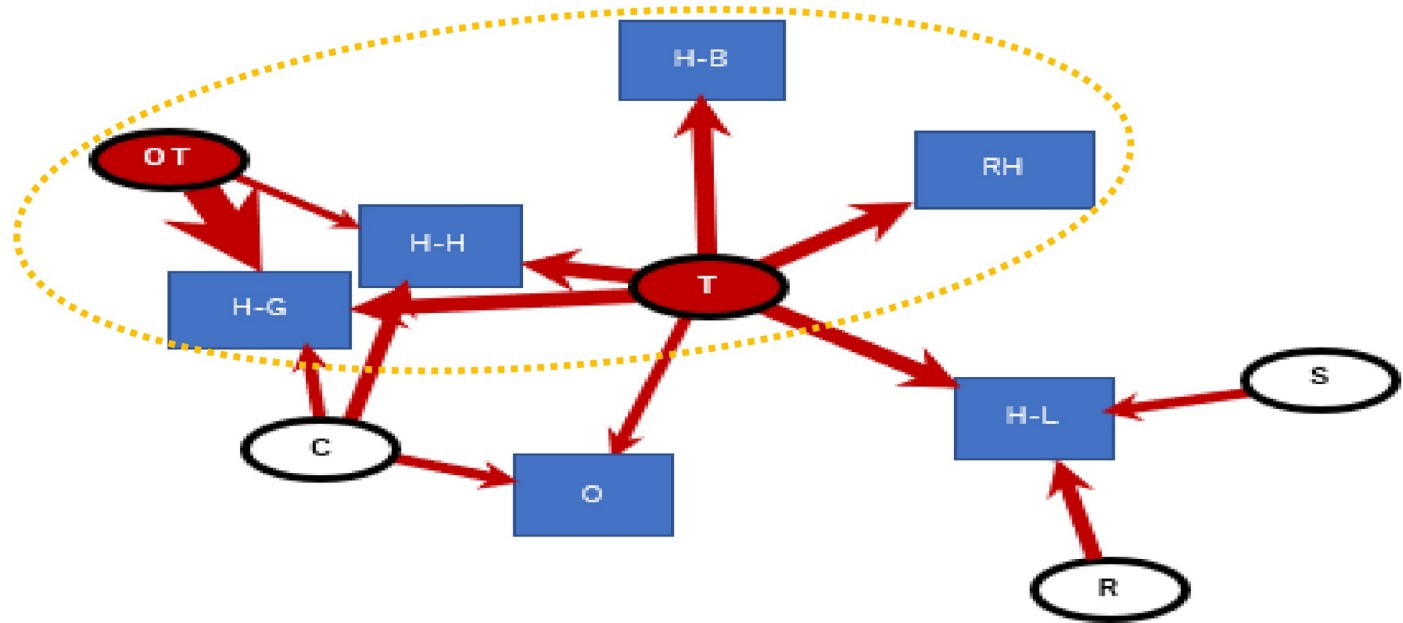

**Fig 2. Causes of serious injuries (T: own tackle, OT: oppose tackle, S: saving, R: rack, C: other collisions, H-H: head to head, H-B: head to opponent's body, RH: head under opponent's body, O: other; the circle was the extracted subgroup by clustering edge betweenness, Modularity Q value = 0.23).**

tackle and the manner of falling after tackle could be related to serious injuries (Fig 2, Modularity Q value = 0.23).

Focusing on the injury and the cause of injury for ASH only, it became clear that a young and inexperienced player tackled and collided his head with the ground or the opponent's body part, or was involved in other collisions. The data on whether the player left the place as soon as possible after the tackle was also extracted (Fig 3, Modularity Q value = 0.37).

## Spinal cord injuries

The spinal injury graphs and the result of the cluster analysis showed that players with spinal cord injuries were older than those with head injuries, but both patient groups had short playing experience. The injuries including VF: vertebral fracture, DS: dislocation of spine, FD: fracture dislocation, S: spinal cord occurred in inexperienced players with 3 years of experience (16E1, 17E2, 18 E3) (Fig 4, Modularity Q value = 0.14)

The causes of the spinal cord injuries include plays under multiple weight pressure, such as scrums, in addition to the one-on-one situation of tackles. It would be necessary to strictly manage the scrum and maul plays in high school players (Fig 5, Modularity Q value = 0.1).

Network analysis revealed that high school players with one or two years of experience tended to suffer from serious head and spinal injuries caused by tackles. Careful consideration for this age would be desired.

## Discussion

The study hypothesis was to clarify the fact that the causes of serious rugby injuries such as head injury and spinal cord injury are due to the years of player experience. That knowledge could help to propose practical rules for injury prevention.

A playing experience of 1 year was defined as the period ranging from 1 to 365 days. First year high school students (16 years old) experience first summer camp in approximately 100

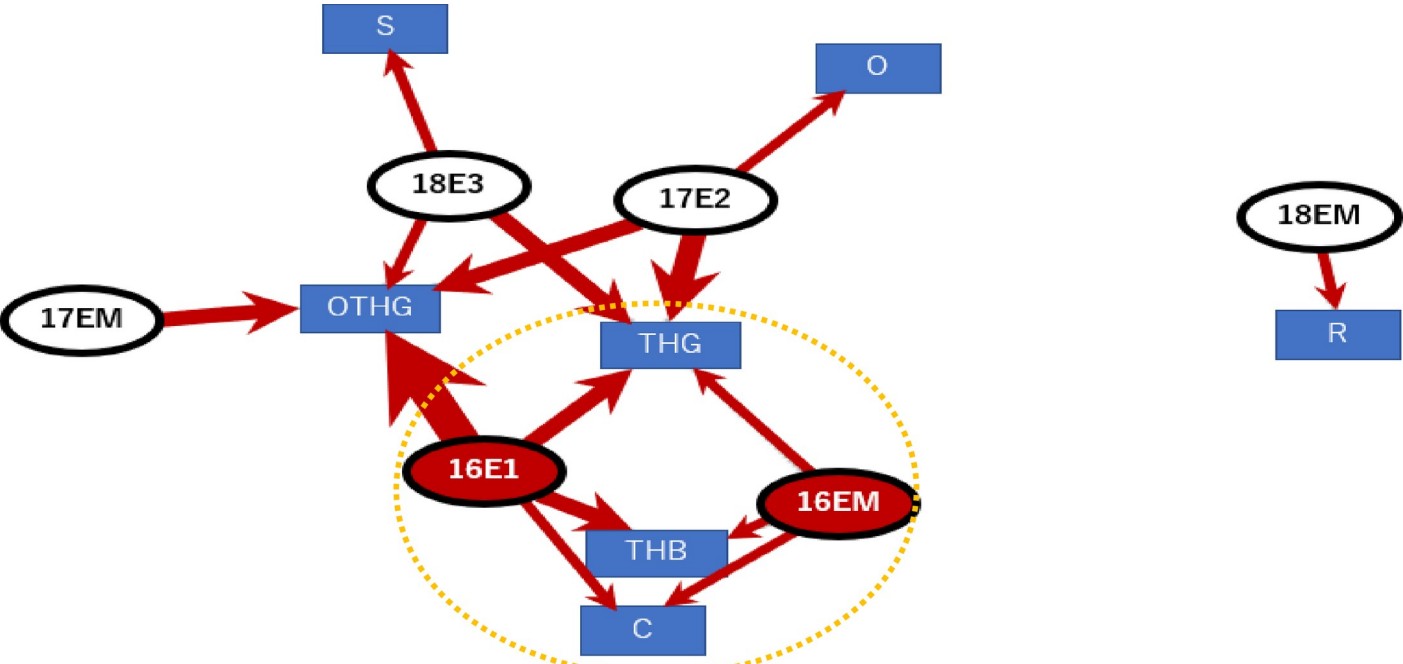

**Fig 3. Injury and cause of injury for acute subdural hematoma only (THG: tackle and head to ground, THB: tackle and head to the opponent's body, OTHG: oppose tackle and head to ground, C: other collision, S: saving, R: rack, O: other; circle was the extracted subgroup by clustering edge betweenness, Modularity Q value = 0.37).**

days (3–4 months) after their starting day. The inexperienced players are forced to face severe physical contacts with opponents with large differences in physique and experience. Many team players who participate in national competitions have more than 10 years of experience. These teams and teams including inexperienced players play against each other during summer camps or national competition qualifiers in the Autumn. There is no doubt that the so-called mismatch situation would be one of the causes of serious injuries [34,35].

Based on the findings of this study, it is an urgent issue to construct special guidelines that would consider the mismatch in cross-age and cross-body size of players. Previous studies on rugby injury also mentioned the difference in physique concerning serious injury [1,2]. Our study suggests that we should consider introducing rules that prohibit head-on collisions in youth inexperienced rugby players. Such specific rules could be developed in collaboration with medical, refereeing, technical and coaching experts. The injury prevention guidelines would also signalize the attention of governments (Japan Sports Agency, Japan Sports Council) and elite rugby organizations (World Rugby) in terms of sports policies that can prevent these serious injuries.

The numbers of female rugby players are also increasing. The empirical study to prevent serious injuries to female players is also urgent. Our study of female rugby players suggested that serious injuries at the lower leg, especially the knee joint that is unique to female athletes, as well as head injuries including concussion [3].

With the aim of acquiring valuable knowledge from data with a graph structure, we performed graph mining [33]. The approaches include frequent pattern detection [36], structure prediction [37], extraction of subgraphs that extract dense nodes (such as subgroup extraction), and calculation of node importance. Among them, it is important to identify closely related communities from network graphs by subgroup extraction and to find highly influential nodes [33].

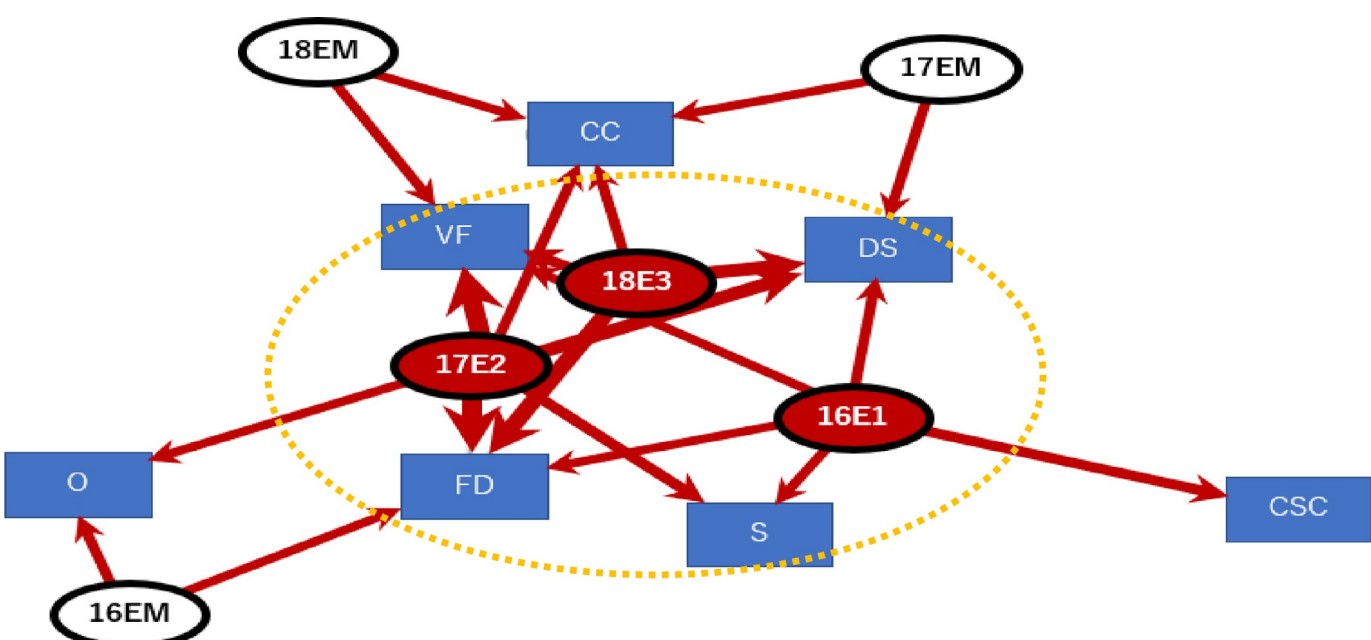

**Fig 4. Spinal injuries and years of playing experience (VF: vertebral fracture, DS: dislocation of spine, FD: fracture dislocation, S: spinal cord, CC: central cord, CSC: cervical spinal cord, O: others; Circle was the extracted subgroup by clustering edge betweenness, Modularity Q value = 0.14).**

The closer the modularity Q value is to 1, the more appropriate the cluster is divided. There were few graphs close to 1 in this study. Whether this is due to the number of data or different factors would be a future task. The value of 0.5 (Fig 1) was a relatively good clustering, which leads to some persuasive considerations. However, even if the modularity values were less than

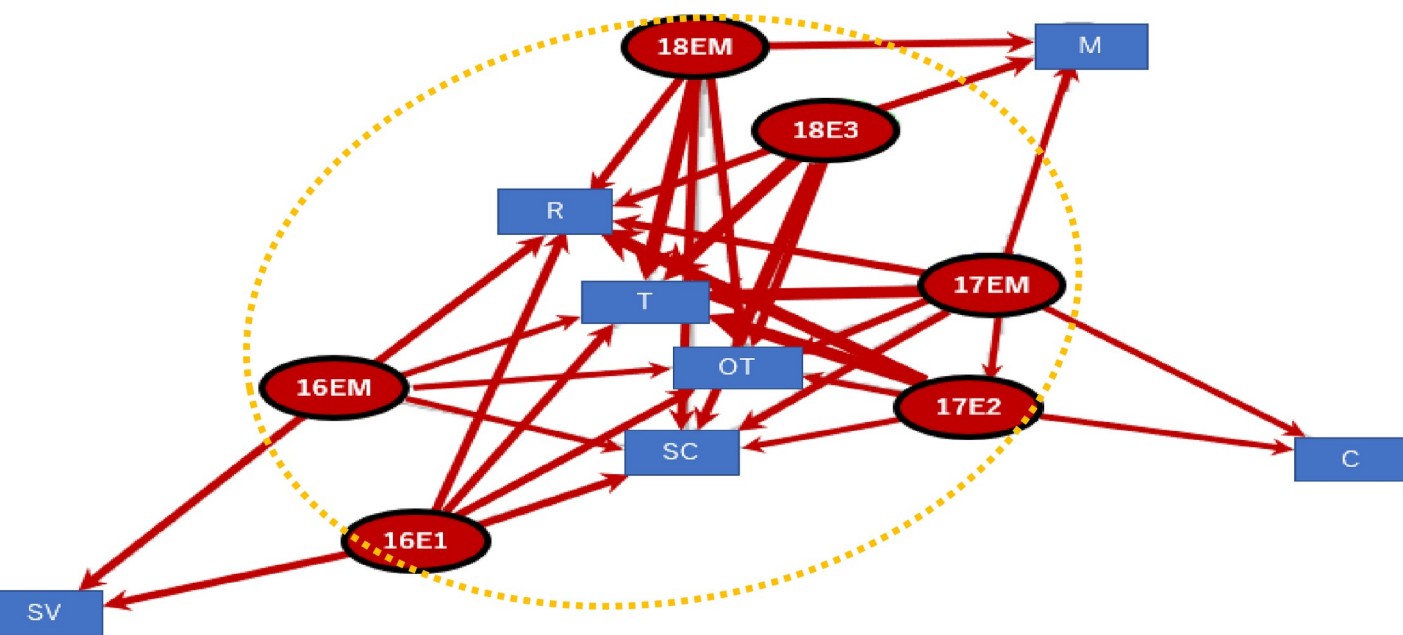

**Fig 5. Years of playing experience and cause of spinal injuries (T: tackle, OT: oppose tackle, SC: scrum, R: rack, M: maul, SV: saving, C: other collision; circle was the extracted subgroup by clustering edge betweenness, Modularity Q value = 0.1).**

0.5, those graphs might be useful findings, suggesting structures of serious injury processes in young rugby players.

This study examined the cause of serious injuries with a bipartite graph. However, in order to consider the relational structure more deeply, multi-stage analysis would be required in the future. In addition, discussion on injury reporting methods in each country seems to be an urgent issue. Serious injury occurs in other ball games and martial arts sports. Cross-sectional study, including specific injury prevention methods, would also be important.

The aim of this study was not to scare young individuals of the possible injuries that could occur when playing rugby because it may be a dangerous sports activity, but it was our goal to design a positive system by sharing information networks structure that for a safer playing environment.

This study has several limitations. First, the small number of data would be a limitation of a one-country study. In the future, collaborative study with foreign countries would be necessary. To unify the data collection methods, it would be necessary to resolve the differences in sports insurance systems in each country. Second, there were causes of injuries that were difficult to determine [34,35], and specifying which situation caused the injury is also difficult to determine. Third, it is also deeply involved with the insurance system. Despite these limitations, our data using more than 14 years of longitudinal data could contribute in preventing serious injuries in rugby players.

## Conclusion

Younger and inexperienced rugby players tended to suffer from serious injuries. Our 14-years (2004–2018) longitudinal data showed that approximately 20 cases of serious injuries occurred in one year. 48 head injuries in 14 years divided 28 subdural hematoma, 5 fracture, 5 acute epidural hematoma, 5 intra cerebral hemorrhage/cerebral contusion, 2 cerebral infarction/cerebrovascular accident, and 3 others. 35 of the 48 (76%) were inexperienced players. 54 Spine cord injuries in 14 years divided 13 fracture dislocation, 13 central cord, 12 spinal cord, 6 cervical spinal cord, 4 vertebral fracture, 4 dislocation of spine, and 2 others. 43 of the 54 (80%) were inexperienced players. Causes of injury were "head to ground", "head to head", "head to leg", "head to the opponent's body part", "head under the opponent's body", and others. The play styles were own tackle, oppose tackle, ruck (a one-on-one situation), other collision not categorized as above, saving (hold the ball on the ground), and other or unknown.

In the bipartite graph of age, years of experience and injury symptoms obtained from 14 years' data, the edge-betweenness centrality network analysis could be an effective method for understanding occurrence structure. Our study suggests that we should consider introducing rules that prohibit head-on collisions in youth rugby.

## Author Contributions

**Conceptualization:** Haruhiko Sato, Akihiko Nakamura, Ichiro Watanabe, Takashi Katsuta, Ichiro Kono.

**Data curation:** Haruhiko Sato, Akihiko Nakamura, Takumi Yamamoto, Ichiro Watanabe.

**Formal analysis:** Takumi Yamamoto.

**Funding acquisition:** Koh Sasaki.

**Investigation:** Takashi Katsuta.

**Methodology:** Koh Sasaki, Akihiko Nakamura, Takumi Yamamoto, Takashi Katsuta.

**Project administration:** Ichiro Kono.

**Supervision:** Ichiro Watanabe.

**Validation:** Koh Sasaki, Akihiko Nakamura, Takumi Yamamoto, Ichiro Kono.

**Visualization:** Koh Sasaki.

**Writing – original draft:** Koh Sasaki, Ichiro Watanabe, Takashi Katsuta, Ichiro Kono.

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
