## [Decision Letter · Decision Letter 0]

6 Jan 2020

PONE-D-19-24936

Who suffers from serious injuries among rugby players? Clarifying the structure of rugby injury through a network edge-betweenness centrality approach

PLOS ONE

Dear Dr Sasaki,

Thank you for submitting your manuscript to PLOS ONE. After careful consideration, we feel that it has merit but does not fully meet PLOS ONE’s publication criteria as it currently stands. Therefore, we invite you to submit a revised version of the manuscript that addresses the points raised during the review process.

Despite the recognized merit of the article, the reviewers suggest some changes mainly in the methods, results, and discussion. I do believe that the article may benefit from a major revision.

We would appreciate receiving your revised manuscript by Feb 20 2020 11:59PM. To enhance the reproducibility of your results, we recommend that if applicable you deposit your laboratory protocols in protocols.io, where a protocol can be assigned its own identifier (DOI) such that it can be cited independently in the future. For instructions see: http://journals.plos.org/plosone/s/submission-guidelines#loc-laboratory-protocols

We look forward to receiving your revised manuscript.

Kind regards,

Filipe Manuel Clemente, PhD

Academic Editor

PLOS ONE

Journal Requirements:

2. Please amend either the abstract on the online submission form (via Edit Submission) or the abstract in the manuscript so that they are identical.

3. Please include your tables as part of your main manuscript and remove the individual files. Please note that supplementary tables should remain as separate "supporting information" files

4. We note you have included a table to which you do not refer in the text of your manuscript. Please ensure that you refer to Table 6 in your text; if accepted, production will need this reference to link the reader to the Table.

5. Please include captions for your Supporting Information files at the end of your manuscript, and update any in-text citations to match accordingly. Please see our Supporting Information guidelines for more information: http://journals.plos.org/plosone/s/supporting-information

Reviewers' comments:

Reviewer's Responses to Questions

**Comments to the Author**

1. Is the manuscript technically sound, and do the data support the conclusions?

Reviewer #1: Partly

Reviewer #2: Yes

Reviewer #3: Yes

2. Has the statistical analysis been performed appropriately and rigorously? 

Reviewer #1: N/A

Reviewer #2: Yes

Reviewer #3: Yes

3. Have the authors made all data underlying the findings in their manuscript fully available?

Reviewer #1: No

Reviewer #2: Yes

Reviewer #3: Yes

4. Is the manuscript presented in an intelligible fashion and written in standard English?

Reviewer #1: No

Reviewer #2: Yes

Reviewer #3: Yes

5. Review Comments to the Author

Reviewer #1: The title is not quite understandable.

The article is very interesting and the use of the network theory could enlightening the interactions of all the variables.

On the other hand, the introduction lacks on focusing the key constructs and does not tackle adequately why using network theory and specifically betweenness centrality. It is too much generalised and focusing on things that does not have relevance for the topic.

The subsection Network Analysis title is repeated in the results, and in the first part it does not explain why using bipartite networks, betweenness centrality (after referring clustering and random walks). This subsection is quite confusing on the relevance of the methods and metrics chosen.

In the section Statistical Analysis, the results does not seem to follow a sequence, repeating the subsection of Network Analysis.

In the Discussion section, there is some information text should be on the method section (e.g. lines 283-285, 300-305, 306).

The conclusion section does not reflect the importance of the study, regarding suggestions and future changes of the rules of the game. Its also confusing on the structure.

Reviewer #2: Congratulations of the paper, due to its relevance, scope, and methodology. However, in order for it to be published I believe it requires two things:

1 - Although the quality of written language is reasonable, it could benefit largely from a revision from a native speaker. The text is understandable, but sometimes difficult to follow. There are also many repetitions and some confusing sentences.

2 - The authors should advance proposals - even if speculative - to circumvent what is occurring with the younger and/or less experienced players. What can be done to improve the situation and make them less exposed to such injuries? Could potential rules changes help? Again, even if speculative, it could provide working ideas for concerning parties, as well as generate interesting and novel prospective studies.

Reviewer #3: The manuscript entitled “Who suffers from serious injuries among rugby players? Clarifying the structure of rugby injury through a network edge-betweenness centrality approach” addresses the usefulness of social network analysis as an effective method for analysing the occurrence structure of rugby league injuries. For achieving this purpose was employed a network centrality analysis of 14-year (2004-2018) longitudinal data in Japan. Thus, a network bipartite graph and subgroup (cluster) analysis were performed to clarify the injured players' experience and the cause of injury. The paper highlights the applicability of social networks as a suitable tool for analysing injuries in rugby league. The topic of the paper is relevant and adequate for this journal. However, some issues must be addressed before the paper can be considered for publication.

The manuscript has the following strong points:

- The topic of the paper is relevant and provides a novel way of using network analysis to analyse injuries in rugby, which can also be extended to other team sports;

- The paper readability is in general good and does not have grammar or spelling mistakes.

The manuscript has the following weak points:

- The abstract is very poor. The authors do not refer the methods and/or statistical procedures used for analysing the data, and do not provide any results;

- The study has no previous established hypotheses;

- The Introduction section is almost (not to say entirely) directed to the properties of the tool/methods (network analysis) and its applications. It is not clear the real relevance of this research if the authors only focus on the tool. The three first paragraphs of the Discussion section have a strong message regarding the mismatch situations that typically occur in sports performance, and it would be highly relevant to re-write the introduction based, not only on methods, but also on this rationale;

- In the methods section, the authors need to provide more details regarding the underlying procedures of a bipartite graph (definition, methods/algorithms, etc). Which social network software, or other tool (please mention the characteristics of the software or tool) was used to create the bipartite network? How was centrality and edge-betweenness centrality calculated? This is important, because, depending on the software (if such a software was used), each one has its own way to calculate centrality-based indicators;

Moreover, why did the authors chose the edge-betweenness centrality and how it is defined/calculated? There is missing information regarding the modularity Q (how the cluster is divided, cut-off points? Software that was used to calculate (Matlab?), etc);

- The Results section is dense. There are lots of tables which can hinder or even distract readers attention. Is there any way that the authors can combine tables in order to reduce their number? For example, the tables depicting the number of injuries, causes, etc, can be amalgamated in only one table;

- The authors do not mention the aim of their study in the first paragraph of the Discussion section, before starting to discuss their results. Moreover, by defining previous hypotheses there is a need to report if they were or not confirmed. Importantly, the authors need to include more references to support their findings;

- The Discussion section can be enriched by providing a more in-depth examination and discussion concerning the mismatch in cross-age and cross-body size of players. Undoubtedly, these encompass two potent messages for the scientific community that can impact in the way that sports organizations and sports federations organize competitions;

- In the conclusion section please report what type of serious injuries do younger and inexperienced players suffer.

I am looking forward to seeing another version of this paper.

6. PLOS authors have the option to publish the peer review history of their article (what does this mean?). If published, this will include your full peer review and any attached files.

Reviewer #1: No

Reviewer #2: Yes: José Afonso

Reviewer #3: No

---

## [Author Response · Author response to Decision Letter 0]

25 Mar 2020

In response to the all reviewer's thankful comments, I have significantly modified, tables, 1_introduction, 2_especially in the methods of network analysis, 3_ results (table amendments), 4_discussions and 5_conclusions.

2020321

---

## [Decision Letter · Decision Letter 1]

25 May 2020

PONE-D-19-24936R1

Clarifying the structure of serious head and spine injury in youth Rugby Union players

PLOS ONE

Dear Dr. Sasaki,

Thank you for submitting your manuscript to PLOS ONE. After careful consideration, we feel that it has merit but does not fully meet PLOS ONE’s publication criteria as it currently stands. Therefore, we invite you to submit a revised version of the manuscript that addresses the points raised during the review process.

Please consider the recommendations of reviewer 3 

We look forward to receiving your revised manuscript.

Kind regards,

Filipe Manuel Clemente, PhD

Academic Editor

PLOS ONE

Reviewers' comments:

Reviewer's Responses to Questions

**Comments to the Author**

1. If the authors have adequately addressed your comments raised in a previous round of review and you feel that this manuscript is now acceptable for publication, you may indicate that here to bypass the “Comments to the Author” section, enter your conflict of interest statement in the “Confidential to Editor” section, and submit your "Accept" recommendation.

Reviewer #2: All comments have been addressed

Reviewer #3: (No Response)

2. Is the manuscript technically sound, and do the data support the conclusions?

Reviewer #2: Yes

Reviewer #3: Yes

3. Has the statistical analysis been performed appropriately and rigorously? 

Reviewer #2: Yes

Reviewer #3: Yes

4. Have the authors made all data underlying the findings in their manuscript fully available?

Reviewer #2: Yes

Reviewer #3: Yes

5. Is the manuscript presented in an intelligible fashion and written in standard English?

Reviewer #2: Yes

Reviewer #3: Yes

6. Review Comments to the Author

Reviewer #2: (No Response)

Reviewer #3: General appreciation

The manuscript entitled “Clarifying the structure of serious head and spine injury in youth Rugby Union players.”, intends to clarify the cause of rugby head and spinal cord injuries through a network centrality analysis of 14-year (2004-2018) longitudinal data in Japan. The authors implemented a network bipartite graph and subgroup (cluster) analysis to examine injured players' experience and the caused of injury. The manuscript is in general well written and provides some interesting points with respect to the aim of the authors. The topic of the manuscript is relevant and adequate for the journal. Regardless, there are some issues that still need to be address before the publication can be considered.

The manuscript has the following strong points:

- The manuscript uses longitudinal data and provides an interesting view of how network analysis can be applied to examine rugby head and spinal cord injuries, which can be further extended to other team sports;

The manuscript has the following weak points:

- In the Introduction section, the authors perform a brief literature review regarding the application of the network analysis in different disciplines (economics, physics, social, etc). However, the authors over-emphasise technical aspects of the network analysis and there is little information concerning its applications on team sports like rugby. What does previous research tells us about the application of the network analysis in rugby? Is there any study that have tried to use social networks to examine injuries? What are the common limitations found in previous studies, beyond an over-emphasis on transversal studies than longitudinal?

- In line 70 of the Introduction section the authors report the importance of network dynamics. What is the difference between a static network and a dynamic network? What about this study? It focuses on static networks or dynamic networks? How does network analysis evaluate dynamic networks?

- The authors report that networks can be used to identify global properties of the network (e.g., structures that emerge more often) as well as local properties (identification of individuals who have a key role in the network). But, what about in this study? Typically, the network analysis conceives the individuals as the network nodes and their relationships are commonly evaluate, for example, through ball-passing actions. In this study, what are the nodes of the network and what type of connections are established among nodes. Although it is implicit throughout the manuscript, the authors need to clarify this. Not all readers are familiar with this type of analysis.

- In the Methods section, there are lots of tables and much information. Is there any way of collapsing information from tables in only two or three tables maximum?

- What are the nodes of the network and what are the links? Is this a static or dynamic network? Why using the edge-betweenness centrality? Is there any limitations of this metric? Please clarify this.

- How were the causes of injuries categorized?

- Please report the aim of the study in the first paragraph of the Discussion section.

- The study hypothesis is very general. Are the authors trying to analyse if the occurrence of serious injuries is related with player experience? Please clarify and make it more specific.

- The authors refer: “Quality of experience”. Is it quality of experience or quantity of experience (i.e. accumulated experience according to the number of years playing rugby)?

- In the Discussion section the authors refer that the data could contribute in preventing serious injuries in rugby players. How? Is the authors aim to signalise the attention of governments and elite rugby organizations in terms of sports policies that can prevent these injuries?

Given the aforementioned, it is my recommendation that the manuscript should not be considered for publication in the current form. However, I am looking forward to see another draft of the paper.

7. PLOS authors have the option to publish the peer review history of their article (what does this mean?). If published, this will include your full peer review and any attached files.

Reviewer #2: No

Reviewer #3: No

---

## [Author Response · Author response to Decision Letter 1]

4 Jun 2020

Dear Reviewer #3 

Appreciate for your kind review. I send same file on the response to review.

with very best regards,

Answre to the reviewer

Q. - In the Introduction section, the authors perform a brief literature review regarding the application of the network analysis in different disciplines (economics, physics, social, etc). However, the authors over-emphasise technical aspects of the network analysis and there is little information concerning its applications on team sports like rugby. What does previous research tells us about the application of the network analysis in rugby? 

We added next study. Sorry it was related soccer (we could not find Rugby analysis studies)

A. It was considered that there was a central player called a hub in the passing behavior research of a soccer game, and there was a power law there. Furthermore, the hub dynamically switched throughout the game. The difference between dynamic networks and static networks is that the former focuses on the variable and diverse play structures occurring in sports games.[18] (page.5 line.72 )

Q. Is there any study that have tried to use social networks to examine injuries? 

We added next study.

A. The factors that play a central role in various physiological parameters during exercise-induced fatigue were clarified by network analysis, and its application to risk management was discussed [13]. (page.4 Line.68 ) 

 Q. What are the common limitations found in previous studies, beyond an over-emphasis on transversal studies than longitudinal? 

A. The reason why there are few longitudinal studies would be that the social rules for ethical use of medical diagnosis and for protecting personal information were not sufficiently established. (page.3 line.44 )　

Q. - In line 70 of the Introduction section the authors report the importance of network dynamics. What is the difference between a static network and a dynamic network?

 A. The difference between dynamic networks and static networks is that the former focuses on the variable and diverse play structures occurring in sports games [18]. (p.5 L:74 )

What about this study? It focuses on static networks or dynamic networks? How does network analysis evaluate dynamic networks? - The authors report that networks can be used to identify global properties of the network (e.g., structures that emerge more often) as well as local properties (identification of individuals who have a key role in the network). But, what about in this study? Typically, the network analysis conceives the individuals as the network nodes and their relationships are commonly evaluate, for example, through ball-passing actions. In this study, what are the nodes of the network and what type of connections are established among nodes. Although it is implicit throughout the manuscript, the authors need to clarify this. Not all readers are familiar with this type of analysis.

 Nodes of network in this study are player groups (age and years of experience), types of injuries, and play styles that cause injuries (tackle, tackled, scrum, etc.). It clarifies the continuity of what kind of players’ experience years, what kind of play, and what part of body collision caused serious injury. These are the dynamic networks which focus on the diverse play structures occurring throughout rugby game. Although this study would clarify a kind of local properties of networks, it might be considered as global properties by promoting international joint study in the future. (p7, L:119)

- In the Methods section, there are lots of tables and much information. Is there any way of collapsing information from tables in only two or three tables maximum?

We change from 9 tables to 4 tables. (p10: table 1,2, p12:Table3, p14:Table4)

- What are the nodes of the network and what are the links? Is this a static or dynamic network? 

Same adobe 

 Nodes of network in this study are player groups (age and years of experience), types of injuries, and play styles that cause injuries (tackle, tackled, scrum, etc.). It clarifies the continuity of what kind of players’ experience years, what kind of play, and what part of body collision caused serious injury. These are the dynamic networks which focus on the diverse play structures occurring throughout rugby game. Although this study would clarify a kind of local properties of networks, it might be considered as global properties by promoting international joint study in the future. (p7. L.119)

Why using the edge-betweenness centrality? Is there any limitations of this metric? Please clarify this. 

A. Edge-betweenness centrality represents the degree of being located in the shortest path bridge between nodes. It was adopted because it would indicate a close relationship among players, injuries, and the causes. The limitation of this metric is that its approximate value may become unstable in case of a large-scale network graph [27, 28]. (p8 L:136)

- How were the causes of injuries categorized? 

A.These classification processes were performed by the safety management committee of the Japanese Rugby Football Union. This committee works closely with the medical committee, coaching committee, referee committee, and technical committee in Japanese Rugby Union, cooperates in classification of the cause of injury based on the detailed play situation reported by the team manager and doctor. (p.7 l.112 ) 

- Please report the aim of the study in the first paragraph of the Discussion section.- The study hypothesis is very general. Are the authors trying to analyse if the occurrence of serious injuries is related with player experience? Please clarify and make it more specific.

We change the sentense under bellow.

A. The study hypothesis was to clarify the fact that the causes of serious rugby injuries such as head injury and spinal cord injury are due to the years of player experience. That knowledge could help to propose practical rules for injury prevention. (p.18 L:300 )

- The authors refer: “Quality of experience”. Is it quality of experience or quantity of experience (i.e. accumulated experience according to the number of years playing rugby)?

We change the sentense

The study hypothesis was to clarify the fact that the causes of serious rugby injuries such as head injury and spinal cord injury are due to the years of player experience. (P18, L:301)

- In the Discussion section the authors refer that the data could contribute in preventing serious injuries in rugby players. How? Is the authors aim to signalise the attention of governments and elite rugby organizations in terms of sports policies that can prevent these injuries? 

We added the sentense under bellow.

Our study suggests that we should consider introducing rules that prohibit head-on collisions in youth inexperienced rugby players. Such specific rules could be developed in collaboration with medical, refereeing, technical and coaching experts. The injury prevention guidelines would also signalize the attention of governments (Japan Sports Agency, Japan Sports Council) and elite rugby organizations (World Rugby) in terms of sports policies that can prevent these serious injuries. (p. 19 L:313 )

---

## [Decision Letter · Decision Letter 2]

9 Jun 2020

Clarifying the structure of serious head and spine injury in youth Rugby Union players

PONE-D-19-24936R2

Dear Dr. Sasaki,

We’re pleased to inform you that your manuscript has been judged scientifically suitable for publication and will be formally accepted for publication once it meets all outstanding technical requirements.

Kind regards,

Filipe Manuel Clemente, PhD

Academic Editor

PLOS ONE

Additional Editor Comments (optional):

Reviewers' comments:

Reviewer's Responses to Questions

**Comments to the Author**

1. If the authors have adequately addressed your comments raised in a previous round of review and you feel that this manuscript is now acceptable for publication, you may indicate that here to bypass the “Comments to the Author” section, enter your conflict of interest statement in the “Confidential to Editor” section, and submit your "Accept" recommendation.

Reviewer #2: All comments have been addressed

Reviewer #3: All comments have been addressed

2. Is the manuscript technically sound, and do the data support the conclusions?

Reviewer #2: Yes

Reviewer #3: Yes

3. Has the statistical analysis been performed appropriately and rigorously? 

Reviewer #2: Yes

Reviewer #3: Yes

4. Have the authors made all data underlying the findings in their manuscript fully available?

Reviewer #2: Yes

Reviewer #3: Yes

5. Is the manuscript presented in an intelligible fashion and written in standard English?

Reviewer #2: Yes

Reviewer #3: Yes

6. Review Comments to the Author

Reviewer #2: I thank the authors for their hard work in improving this manuscript. I believe this updated version constitutes a major improvement over previous versions and will provide a solid and relevant contribution for the field of Sports Sciences.

Reviewer #3: (No Response)

7. PLOS authors have the option to publish the peer review history of their article (what does this mean?). If published, this will include your full peer review and any attached files.

Reviewer #2: No

Reviewer #3: No

---

## [Editor Report · Acceptance letter]

11 Jun 2020

PONE-D-19-24936R2 

Clarifying the structure of serious head and spine injury in youth Rugby Union players 

Dear Dr. Sasaki:

I'm pleased to inform you that your manuscript has been deemed suitable for publication in PLOS ONE. Congratulations! Your manuscript is now with our production department. 

Kind regards, 

on behalf of

Dr. Filipe Manuel Clemente 

Academic Editor

PLOS ONE